# French Survey on Pain Perception and Management in Patients with Locked-In Syndrome

**DOI:** 10.3390/diagnostics12030769

**Published:** 2022-03-21

**Authors:** Estelle A. C. Bonin, Zoé Delsemme, Véronique Blandin, Naji L. Alnagger, Aurore Thibaut, Marie-Elisabeth Faymonville, Steven Laureys, Audrey Vanhaudenhuyse, Olivia Gosseries

**Affiliations:** 1Coma Science Group, GIGA-Consciousness, University of Liège, 4000 Liege, Belgium; estelle.bonin@uliege.be (E.A.C.B.); zoe.delsemme@mil.be (Z.D.); n.alnagger@uliege.be (N.L.A.); athibaut@uliege.be (A.T.); steven.laureys@uliege.be (S.L.); 2Centre du Cerveau^2^, University Hospital of Liège, 4000 Liege, Belgium; 3Association des Locked-in Syndrome (ALIS), 92100 Boulogne-Billancourt, France; veronique.blandin@alis-asso.fr; 4Centre Interdisciplinaire d’Algologie, University Hospital of Liège, 4000 Liege, Belgium; mfaymonville@chu.ulg.ac.be; 5Sensation and Perception Research Group, GIGA-Consciousness, University of Liège, 4000 Liege, Belgium; 6Joint International Research Unit on Consciousness, CERVO Brain Research Centre, CIUSS, University Laval, Quebec City, QC G1V 0A6, Canada

**Keywords:** survey, locked-in syndrome, pain, quality of life, guidelines

## Abstract

Patients with locked-in syndrome (LIS) may suffer from pain, which can significantly affect their daily life and well-being. In this study, we aim to investigate the presence and the management of pain in LIS patients. Fifty-one participants completed a survey collecting socio-demographic information and detailed reports regarding pain perception and management (type and frequency of pain, daily impact of pain, treatments). Almost half of the LIS patients reported experiencing pain (49%) that affected their quality of life, sleep and cognition. The majority of these patients reported that they did not communicate their pain to clinical staff. Out of the 25 patients reporting pain, 18 (72%) received treatment (60% pharmacological, 12% non-pharmacological) and described the treatment efficacy as ‘moderate’. In addition, 14 (56%) patients were willing to try other non-pharmacological treatments, such as hypnosis or meditation. This study provides a comprehensive characterization of pain perception in LIS patients and highlights the lack of guidelines for pain detection and its management. This is especially pertinent given that pain affects diagnoses, by either inducing fatigue or by using pharmacological treatments that modulate the levels of wakefulness and concentration of such patients.

## 1. Introduction

Locked-in syndrome (LIS) is a neurological disorder that occurs after a brainstem lesion, most commonly due to a stroke [1,2]. It results in paralysis of all four limbs, head, and face (quadriplegia or quadriparesis) and impaired speech (aphonia or hypophonia). LIS patients are conscious, have preserved cognitive functions, and usually communicating through eye movements and blinking [3]. Also, despite their quadriplegia, LIS patients retain tactile sensitivity [4]. The treating physician may erroneously diagnose the LIS patient as being in a coma, or as occupying one of several disorders of consciousness (DoC). These include the minimally conscious state (i.e., with discernible but fluctuating evidence of consciousness without effective communication; MCS [5,6]) or the unresponsive wakefulness syndrome (i.e., eye opening period but no sign of consciousness; UWS [7]) [8,9].

Once a LIS patient becomes medically stable and receives appropriate medical care, life expectancy increases by several decades [2]. Therefore, the elapsed time between brain injury and LIS diagnosis is precious, and finding the appropriate treatment both efficiently and effectively is of the upmost importance. A LIS diagnosis usually takes between 2.5 months and 4–6 years, and in the majority of the cases (55%) diagnosis is made by the patient’s relative, not the treating physicians (23%) [10]. Pain management using pharmacological treatments in post-comatose patients can have deleterious side effects, such as increasing fatigue or decreasing awareness. This can lead to misdiagnosis and have important consequences on the continuation of care. The International Association for the Study of Pain (IASP) defines pain as “an unpleasant sensory and emotional experience associated with, or resembling that associated with, actual or potential tissue damage” [8,9,11]. LIS patients can suffer from acute pain (i.e., which usually occurs suddenly, most often due to inflammation or severe medical condition [11]) and/or chronic pain (i.e., pain lasting longer than 3 months or beyond the expected period of healing of tissue pathology) [12]. They can also suffer from neuropathic pain, which is defined as a “pain arising as a direct consequence of a lesion or disease affecting the somatosensory system” [13]. In addition, these patients are likely to develop spasticity, the severity of which correlates with pain intensity [14]. Pain perception has been previously investigated in DoC patients. Neuroimaging studies have shown that painful stimulations applied to MCS patients can lead to the activation of brain regions involved in the sensorial and affective processing of pain (i.e., insula, anterior cingulate cortex and secondary somatosensory cortex), similar to observations in healthy subjects [15,16]. Meanwhile, in UWS patients, only the primary somatosensory cortex (i.e., involved in the sensory processing of pain) was activated following a painful stimulation, suggesting that pain processing in UWS could be compromised [17,18].

There are minimal scientific studies investigating the processing of pain in LIS. A survey conducted in 2008 showed that despite the fact that 46% of the LIS patients reported moderate or extreme pain, 72% of patients reported a good quality of life [19]. Another study showed that pain was associated with a decrease of quality of life in post-stroke patients [20]. Furthermore, it has been shown that patients’ mental health and the presence of physical pain correlate with the frequency of suicidal thoughts, suggesting that pain has a significant impact on the quality of life of these patients as in other diseases like cancer, fibromyalgia syndrome or stroke [2,20,21,22,23,24].

At present, no study in LIS patients provides a detailed description of pain and related issues (frequency, location, treatment protocols, etc.). In this survey, we investigated the presence and management of pain in LIS patients. The first aim was to re-evaluate the results obtained in previous studies regarding the presence of pain and the impact on quality of life. Indeed, as LIS patients often suffer from spasticity and long-term immobility, we hypothesize that the presence of chronic pain in these patients is frequent. We also hypothesize that the presence of pain will have a negative impact on their quality of life via affecting their cognition/mental abilities, emotional regulation and sleep quality. The second aim of this survey was to further characterize the pain experienced by this population (type and frequency of pain, means of communicating the pain, type of treatment used, degree of satisfaction with treatments, etc.). We expect that the majority of the patients will have used pharmacological treatments, with a minority who will have used non-pharmacological treatments.

## 2. Materials and Methods

This survey was sent by email to LIS members of the Association du Locked-In Syndrome (ALIS, Paris, France). The survey was available in French, with a paper version and an online version. The study conformed to the principles of the Declaration of Helsinki and French Good Clinical Practices. According to the French law (Article L1121-1, Law n°2011–2012 29 December 2011—art. 5) and the Belgian law (law of 7 May 2004), online surveys are not covered by the law on human experimentation. Therefore, ethical approval was not needed. Completion of the questionnaire was non-remunerated, voluntary, and anonymous. Completion was considered as consent for participation in the survey.

The questionnaire follows the CHERRIES checklist guidelines (see Appendix A). The survey was developed on the basis of previous surveys carried out among LIS patients studying their well-being and quality of life [19,25], and tested internally by the investigators before sending it to the participants. The online survey was done using a Google form and a paper version was also made available (PDF). The questionnaire was composed of 28 questions divided into two main sections: a first part collecting respondents’ socio-demographic information and clinical status (i.e., age, sex, time since injury, etiology, own of an electric wheelchair, presence of tracheotomy and gastrostomy, use of verbal or code communication; in compliance with the GDPR), and a second part consisting of multiple select and closed questions on pain perception and management (e.g., type and frequency of pain, daily pain impact, tested treatments) (Appendix A). Participants were also asked how they had completed the questionnaire (i.e., alone, with the help of a family member or health care professional).

For the first question, participants were asked to specify whether they had suffered from pain within the two weeks prior the survey completion. Only participants who had experienced pain within the last two weeks could continue with the questionnaire. Therefore, for patients who had not experienced pain, only socio-demographic data and clinical status were collected. Question 2 focused on the localization of pain. In questions 3 to 5, a visual analogue scale (VAS) ranging from 0 to 10 was used to assess pain intensity (0 = no pain, 10 = most intense pain). Questions 6 to 9 were based on the DN4 questionnaire [26]. The DN4 is a validated and simplified questionnaire administered by clinicians to detect neuropathic pain. Each question consists of a number of items (10 in total) and the participant is asked to indicate the presence or absence of each item by ticking “yes” or “no”. An item that is present will have a score of 1 while an item that is absent will have a score of 0. The sum of all 10 items corresponds to the total score, and the threshold value for the diagnosis of neuropathic pain is 4/10. If patients show a score greater than or equal to 4/10, then they are considered to be suffering from neuropathic pain. In question 10, participants were asked to indicate how long they had been experiencing pain (if more than 3 months it could be considered as chronic pain). Questions 11 and 13 queried their experiences of pain prior to LIS. Participants who had experienced pain before LIS (Question 11) were asked to evaluate the pain intensity using the VAS (Question 12) and describe the evolution of this pain after LIS (Question 13). For question 14, patients were asked to indicate whether their pain was continuous (i.e., present all the time) or discrete (i.e., present at certain times of the day). For question 15, participants were asked to indicate how they expressed their pain. Questions 16 and 17 focused on elements that alter the experience of current pain: mood/emotion (i.e., negative or positive valence), temperature (negative or positive), position whilst sitting, type of care environment (e.g., nursing, physiotherapy), touching the painful area, fatigue, engaging in physical exercises and available equipment (e.g., types of cushioning, using a wheelchair). Questions 18 to 22 were about the influence of pain on cognitive abilities, sleep quality and emotional regulation. For questions 19 and 22, patients used the VAS ranging from 0 to 10 to assess the influence of pain on their quality of sleep and emotional regulation (0 = no influence, 10 = strong influence). For questions 23 to 27, participants were asked to indicate whether they were taking pharmacological and/or non-pharmacological treatments for their pain, and if so to list them. The different pharmacological treatments were then classified into nine categories based on the WHO classification. Level 1 painkillers denoted non-opioids, level 2 painkillers referred to weak opioids and level 3 painkillers were strong opioids [27]. For questions 25 and 27, patients used the VAS ranging from 0 to 10 to assess the efficiency of these treatments (0 = not efficient, 10 = strongly efficient). Finally, for question 28, they were asked if they were willing to test new pharmacological or non-pharmacological treatments.

Data were exported and analyzed from the Google form in .csv format. Data collected from the paper version were added to these files. Statistical analyses were performed using R studio (version 4.0.2) software. For the descriptive analyses, we used subject counts and percentages to describe the categorical responses and means ± standard deviation (SD) when assessing pain intensity/influence or treatment efficiency. Regarding questions with multiple select answers, we based the calculation on the percentage of the number of participants who replied to each question. As the variables were categorical and/or ordinal variables, non-parametric statistics (i.e., Pearson’s chi-squared test, Fischer test and Wilcoxon rank sum test) were performed.

## 3. Results

### 3.1. Demographic Information and Clinical Status

From the 300 patients contacted by email, 59 surveys were collected, and 8 were excluded from the analysis as there were incomplete data on pain. A total of 51 participants (41 from the online version and 10 from the paper version) were included in the analysis, representing a 17% acquisition rate. Seven participants (14%) completed the survey by themselves, 10 (20%) needed the help of a family member, eight (16%) needed the help of a health care professional (i.e., two psychologists, three occupational therapists, one nurse, one social worker and one ALIS member) and 26 (51%) did not answer to this question (i.e., which corresponds to patients who did not experience pain). Figure 1 provides an illustrative summary of the results.

Among the respondents that declared their sex, there was the same number of women as men (n = 18). Fifteen participants did not specify their sex. The mean age was 51 ± 12 y.o. (range from 22 to 81 y.o.), and the mean time since injury was 11 ± 8 years (range from 1 to 36 years). Table 1 summarizes the socio-demographic information and clinical status of the 51 patients (for individual data see Appendix A).

### 3.2. Past and Current Pain

Half of the participants answered that they had experienced pain at some point during the last two weeks (25/51, 49%; median pain intensity: 6/10). The 26 participants who did not feel any pain did not complete the rest of the questionnaire. Pearson’s chi-squared test and fisher tests were performed to see if there were any differences between the painful and the non-painful group according to sex, etiology, presence of tracheotomy/gastrostomy. For the time since injury, a Wilcoxon rank sum test was used to investigate the differences between the two groups. No significant difference between these different categories was found between the two groups (Table 1). Results regarding the localization and communication method are summarized in Figure 2 (for individual details see Appendix A).

Questions 6 to 9 were based on the DN4 questionnaire. Regarding the features of the pain, 10 participants reported electrical shock-like sensations (40%), six experienced burning (24%), four experienced painful cold sensations (16%), and 10 reported none of these proposed features (40%). When participants were asked if they had experienced any other symptoms in the painful area, 12 participants (48%) had a vice-like pressurized feeling, six (24%) experienced sensation akin to pin-pricks, six (24%) felt a numbing sensation, five (20%) felt a tingling sensation, three (12%) had an itching sensation, and nine (36%) had none of the proposed symptoms. Twenty participants (80%) answered that they felt a decrease in touch sensitivity while sitting, three (12%) felt a decrease in touch sensitivity whilst being touched, and two (8%) did not feel any decrease in touch sensitivity. Seventeen participants (68%) reported that their pain was not caused or altered by friction. From these questions, we identified nine participants out of 25 (36%) with neuropathic pain (i.e., total score > 4).

Participants were then asked to indicate when they started to feel these painful sensations. Twelve participants out of 25 felt these pains for more than one year (48%), five between 6 months and one year (20%), six between 3 and 6 months (24%) and two between 1 and 3 months (8%). As pain is considered chronic when it lasts for more than three months, almost all the participants suffered from chronic pain (23/25; 92%). The majority of the participants claimed not to have experienced any pain before the LIS (21/25; 84%), whereas some participants reported already experiencing pain prior to their brain injury (2/25; 8%) in addition to some participants who were unsure of whether or not they experienced pain prior to their brain injury (2/25; 8%). Finally, two participants noted a decrease in this pain following the LIS and one did not feel any change.

When describing the current pain experienced, 21 participants out of 25 reported having discrete episodes of pain (84%) and four reported continuous pain (16%). Four participants out of 21 felt discrete episodes of pain less than once a day (19%), two once a day (10%), three more than once a day (14%) and 12 did not know (57%). Six participants out of 21 felt discrete episodes of pain during the evening (29%), five during the morning (24%), five during the afternoon (24%) and five could not give an estimation of the time of painful experiences (24%).

### 3.3. Factors That Influence Pain

Participants were asked to select from different elements of daily life that could increase or decrease their pain levels (Figure 3). It should be noted that eight (32%) participants reported that certain elements such as type of care (3/25; 12%), temperature (2/25; 8%), touching (2/25; 8%), tiredness (2/25; 8%), supine (2/25; 8%) and sitting (1/25; 4%) positions could both decrease and increase pain.

Regarding the effects of pain on cognition/mental abilities, emotional regulation and sleep, results are summarized in Figure 4. Fisher tests were performed to see if there was any relationship between the use of pain treatments and the presence of sleep disturbance or cognitive disabilities. No link could be established between the presence/absence of pain treatment and the presence/absence of sleep or cognitive disturbances (Appendix A).

### 3.4. Treatments

Details of the use and efficacy of pharmacological and non-pharmacological treatments are summarized in Figure 5. Fisher tests were performed to see if there was any relationship between pain intensity (VAS score greater or equal to 5 compared to VAS score lower than 5) and the use of pain treatments. No significant link could be established between pain intensity and the presence/absence of pain treatments (Appendix A).

## 4. Discussion

The results of the survey show that half of the participants had experienced pain during the two weeks before the completion of the questionnaire. The other half of the participants did not report suffering from pain during that time, suggesting that an effective pain management plan was in place. Among the participants who reported pain, 92% suffered from chronic pain. A large proportion of the participants reported that their sleep and cognition/mental abilities were affected by their pain. These two findings confirm our first hypotheses and previous results obtained in former studies [19,20,21]. The second purpose of this observational study was to describe in more detail the characteristics, assessment and management of pain in LIS patients by directly interviewing the persons concerned. Importantly, more than half of the participants did not communicate their pain with clinical staff, which raises questions about the proper detection and management of pain. Regarding the treatments employed, painkiller level 1 (non-opioids) was the most commonly used (73%). Only a minority of participants had tried non-pharmacological treatments such as osteopathy, acupuncture and electromagnetic therapy, yet more than half were willing to try such options.

### 4.1. Past and Current Pain

Half of the participants interviewed in this study reported pain (49%). The vast majority of the participants reported pain in the lower limbs (84%), followed by headaches (28%) and pain in the upper limbs (24%). It can be assumed that the pain in the limbs could be related to the fact that LIS patients remain completely immobile and are unable to move by themselves (71% of the participants were in wheelchairs).

Regarding the type of pain experienced by the participants, the majority occurred after the brain injury therefore being more likely to be a direct result of the patient’s condition. Moreover, 92% had chronic pain, 84% reported discrete episodes of pain, including 19% who experienced pain more than once a day. Neuropathic pain is a category of chronic pain, but not all chronic pain conditions are neuropathic. In our study, 36% of the respondents had neuropathic pain [26]. The characteristics of neuropathic pain that stand out the most were sensations of electric shocks (40%), a vice-like pressurized feeling (48%) and decrease of touch sensitivity while sitting (80%). Out of the nine participants identified as having neuropathic pain, seven were in LIS following a stroke. Several studies have ventured to develop treatments for post-stroke neuropathic pain, including by employing motor cortex stimulation. However, the results have yet to be replicated in LIS patients [28].

Our study also highlights that 52% of participants experiencing pain did not communicate it to others. Of those who communicated their pain, 68% did so using an alphabetical code. Forty-four percent of the participants stated that they expressed their pain through crying. Notably, other means of communication (e.g., wincing) are not always easy to detect or may be confounded with reflexive behavior. Only 28% of the participants used a communication code to communicate their pain. This could be explained by the fact that, in the case of acute pain in particular, using an alphabetical code requires significant concentration, which can be tiring and discouraging for the patient. The development of new means of communication such as the use of a brain–computer interface requiring less cognitive effort could be interesting for these patients who do not communicate their pain [29].

### 4.2. Factors That Influence Pain

Interestingly, we found a number of factors that can either increase or decrease perceived pain depending on the individual. For example, in some patients, supine and sitting position can increase or decrease pain depending on the patient. This exemplifies the high variability between subjects, pain is subjective and therefore the treatment and management of pain approached on a case-by-case basis. Importantly, 36% of patients reported that tiredness worsened their pain and 40% claimed that pain disrupted their sleep. In this sense, tiredness, sleep quality and pain perception may constitute a feedback cycle of pain-influencing factors. There are several studies that support this proposed bidirectional relationship, for example one study indicated that 50–70% of patients with chronic pain suffer from sleep disorders [25]. Furthermore, patients who suffer from poor sleep quality during the proceeding night have an increase in pain during the following day. Conversely, a significantly painful day has been shown to be followed by subsequent sleep disturbances the following night [30]. It is therefore important to find a balance between the management of these pains and the impact of such treatments on their levels of arousal and quality of sleep.

Regarding the deleterious effects of pain on mental health, cognitive abilities and emotional regulation, our results are in line with previous surveys. One survey shows that 55% of LIS patients reported experiencing significant anxiety and/or other comorbid mood disorders, including 13% of LIS patients reporting being depressed [20] and 27% reporting experiencing suicidal thoughts [25].

Additionally, previous research has reported that perceived pain and life satisfaction are inversely correlated [10,21]. Exploring this, some authors have highlighted a number of variables associated with life satisfaction of LIS patients. These include a lack of mobility concerning recreational activities and those within the community and particularly the non-recovery of speech production [19]. Preserved communication is likely to significantly improve the quality of life of LIS patients. Indeed, a previous survey associated LIS patients reporting a good quality of life with the ability to produce speech and lower subsequent rates of anxiety [5].

### 4.3. Treatments

Our results show that most LIS patients use pharmacological treatments, in particular level 1 painkillers (73%) to reduce perceived pain. In general, the subjects seem to be moderately satisfied with pharmacological treatments, but it is important to point out that, like pain sensitivity, there is a large inter-individual variability. In addition, the treatments may cause several deleterious side effects, such as increased fatigue and cognitive diminution. These can deteriorate their quality of life and impact their diagnoses. Indeed, LIS patients are confronted daily with substantial cognitive and attentional demands since their only means of interaction with the outside world is via blinking, eye movements or residual finger movements. Consequently, the attentional resources required to master the communication technology are significant.

Avoiding the undesirable effects of heavy analgesic medication may facilitate more efficient communication by allowing the patients to take advantage of all communication tools at their disposal. Therefore, the improvements in comfort, potentially combined with a reduction in medication, will significantly improve the quality of life of these patients.

In our survey, few participants (12%) tried osteopathy, acupuncture and electromagnetic therapy as complementary/non-pharmacological treatments. Researchers have also demonstrated the effectiveness of physiotherapy (combined with other treatments) in the management of LIS patients [31]. Surprisingly none of the participants who reported pain had ever tried complementary approaches such as hypnosis, relaxation or meditation. These complementary techniques have yielded positive effects in other patient populations suffering from acute or chronic pain [32]. Neuroimaging studies carried out in meditation experts have shown that a decrease in pain sensitivity was associated with decreases in brain activity in brain regions involved in emotional processing and executive functions, in conjunction with increases of brain activity in regions involved in pain processing. This decrease in the cognitive and emotional control of pain could facilitate the an alteration in the processing of pain as a neutral stimulus rather than an unpleasant one [33,34]. Regarding the use of hypnosis to modulate pain, studies have shown a decrease of brain activation in areas involved in sensory and affective processing of pain during hypnosis [35,36]. This technique has also been shown to be effective in reducing pain in patients with chronic pain [37,38]. Therefore, it would be relevant to further investigate the potential benefit of hypnosis and meditation in the management of LIS patients. Proposing a global non-pharmacological approach could help reducing pain while preserving patient’s level of arousal.

Another important factor in the management of LIS is the speed of finding appropriate treatment. Previous studies confirmed that there is a reduction in mortality and improvements in functional recovery in the case of early and intensive rehabilitation (management within approximately 1 month after the morbid event) [19,39].

### 4.4. Limitations and Future Directions

Our study has several caveats that limit the generalizability of our results. Firstly, the sample size is small (n = 51 in total but only 25 who completed the entire questionnaire). It would be interesting to follow-up with a larger representative sample. Unfortunately, as LIS is a rare pathology, gathering a large sample remains a challenging consideration for future studies. Nevertheless, the vast majority of studies in the literature on LIS patients are case studies and published group studies are approximately the same size as our sample size [19,40,41,42,43,44,45,46]. The acquisition rate however was low (17%) but this should be put into perspective with the context in which the study was carried out and the type of population targeted. Indeed, we do not know how many participants did actually read the email; some LIS patients may not have had the tools and help to fill the questionnaire; and ALIS sends several questionnaires per year, which can be discouraging for the LIS patients because it requires time and effort. However, to our knowledge, this is the first survey that directly addresses LIS patients regarding pain management. Secondly, only patients who had pain in the two weeks prior the completion of the survey were included. Longitudinal data are required to see if there is a correlation between the presence of these painful sensations and the diagnosis of these patients (i.e., a patient whose attention is diminished because of pain could be misdiagnosed as UWS). Thirdly, some questions lack precision, especially regarding the type of care, the influence of temperature or the dosage of pain treatments. In particular, it would have been interesting to know the dose of pain treatments, since it can impact the diagnosis of the patient, their sleep quality and cognitive abilities. Indeed, the use of a too high dose of painkillers can lead to an increase in tiredness and thus a decrease in the level of arousal and communication abilities. Interventional studies testing the effects of specific painkillers should be conducted. Additionally, future studies should use alternative response format (e.g., Likert scales) to collect more detailed data than binary format (e.g., yes/no).

## 5. Conclusions

This survey is one of the first studies that provides an overview of pain and its management in LIS by directly interviewing patients. This study highlights that (1) half of LIS patients experience pain, in particular chronic pain. (2) LIS patients do not communicate often about their pain perception, which does not facilitate good management. This means that caregivers must be vigilant about detecting signs of pain. The systematic implementation by clinical teams of a specific communication code for pain could help detect these signs more quickly. (3) Pain in LIS patients influences quality of life and, more specifically, sleep and emotion. This will have an impact on patients’ diagnosis by affecting their level of concentration and motivation. (4) Pharmacological treatments are still the mainstay of pain management for these patients. Such pharmacological treatments can lead to various side effects, including drowsiness, increased fatigue, and cognitive slowing. An exciting other option to reduce pain and avert these side effects would be to use non-pharmacological treatments, but this requires further investigation in future studies.

## Figures and Tables

**Figure 1 diagnostics-12-00769-f001:**
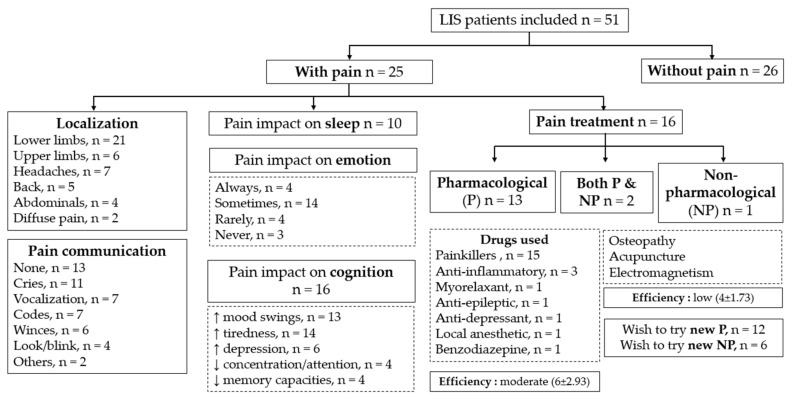
Summary of main results. (LIS = locked-in syndrome, P = pharmacological treatment, NP = non-pharmacological treatment).

**Figure 2 diagnostics-12-00769-f002:**
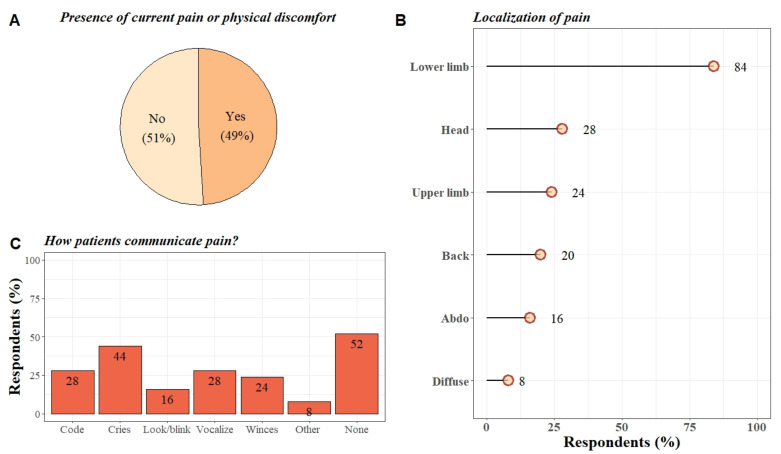
Summary of the results on past and current pain. (**A**) Pie chart of the distribution (in percentages) of LIS patients who had experienced pain/physical discomfort versus LIS patients who had not experienced pain/physical discomfort in the last two weeks before the study. (**B**) Lollipop graph representing the distribution (in percentages) of the different areas of the body reported as painful for these participants (multiple select answer). (**C**) Bar plot showing the distribution (in percentages) of the different means of communication used by the participants to express their pain. (Other = communication methods such as verbalization via a speech valve and the presence of acute spasticity), (Code = Use of a communication code).

**Figure 3 diagnostics-12-00769-f003:**
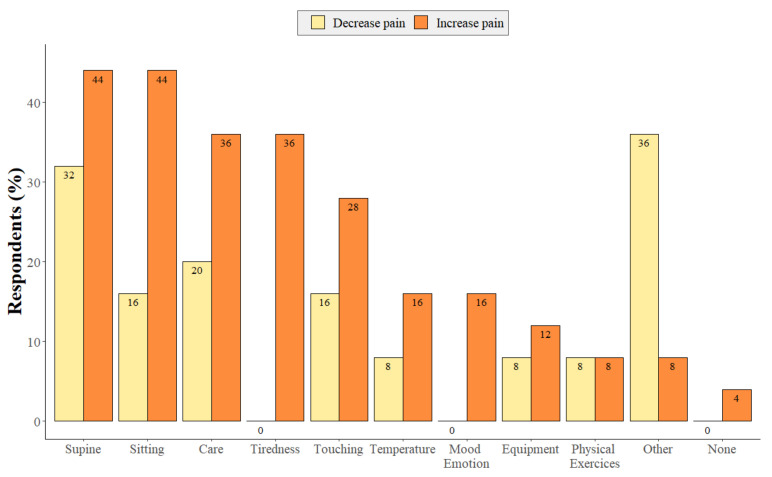
Distribution of the different elements that can increase (in orange) or decrease (in yellow) pain in LIS patients (multiple select answer). (Other = feeding, daily handling increase pain, Botox injection, use of medication decreases pain).

**Figure 4 diagnostics-12-00769-f004:**
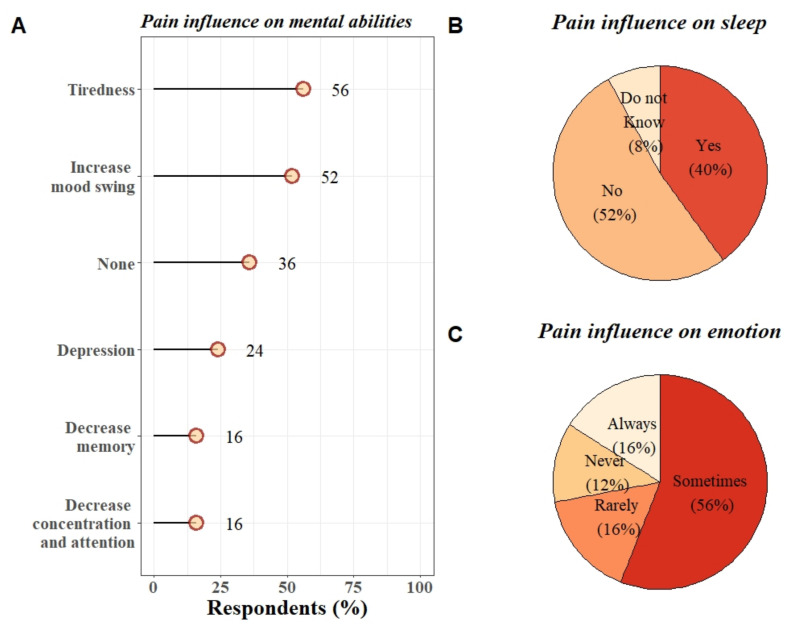
Influence of pain on (**A**) mental abilities (multiple select answers), (**B**) sleep, and (**C**) emotions in LIS patients.

**Figure 5 diagnostics-12-00769-f005:**
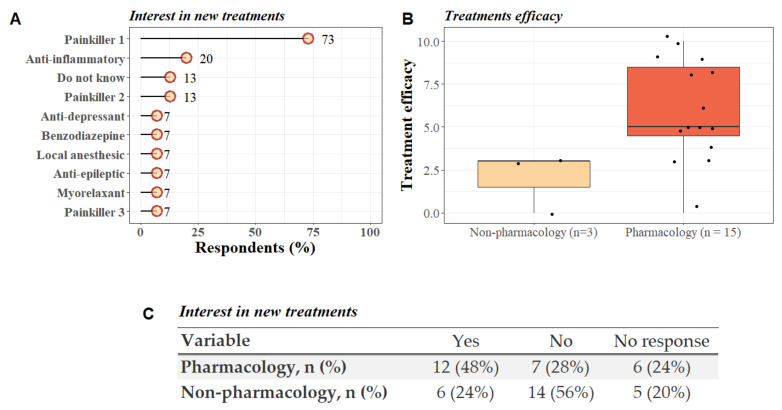
Pain treatment in LIS patients. (**A**) Lollipop plot representing the type of pharmacological treatments used by patients to treat pain (based on the WHO classification [27]). More than half of the patients used at least two types of medication (5/15; 53%). (**B**) Boxplot representing the treatments efficacy according to the participants opinion (0 = not effective, 10 = very efficient) and the type of treatments used (pharmacological vs. non-pharmacological). Additional information: only three out of 25 (12%) participants reported taking non-pharmacological treatments (osteopathy, acupuncture and electromagnetic therapy), among these three, two patients were also using pharmacological treatments to manage pain. (**C**) Table shows the patients’ opinion about willing to try new pharmacological and/or non-pharmacological treatments to prevent pain.

**Table 1 diagnostics-12-00769-t001:** Summary of the socio-demographic information and clinical status for the whole sample and for the group of patients with and without pain.

Variable	Total Sample	Patients with Pain	Patients without Pain	*p*-Value ^1^
	(n = 51)	(n = 25)	(n = 26)	
**Sex, n (%)**				0.31
Female	18 (50%)	10 (62%)	8 (40%)	
Male	18 (50%)	6 (38%)	12 (60%)	
Unknown	15	9	6	
**Etiology, n (%)**				0.57
Stroke	41 (80%)	21 (84%)	20 (77%)	
TBI	4 (7.8%)	2 (8.0%)	2 (7.7%)	
Infection	2 (3.9%)	0 (0%)	2 (7.7%)	
Other	4 (7.8%)	2 (8.0%)	2 (7.7%)	
**Time since injury, Median (IQR)**	9 (6–18)	6 (3–18)	10 (6–16)	0.26
**Tracheotomy, n (%)**	26 (51%)	15 (29%)	1 (22%)	0.21
**Gastrostomy, n (%)**	34 (67%)	19 (37%)	15 (29%)	0.17
**Verbal communication, n (%)**	13 (25%)	5 (9.8%)	8 (16%)	0.38
**Use of an alphabetic code, n (%)**	35 (69%)	19 (37%)	16 (31%)	0.27
**Own a wheelchair, n (%)**	36 (71%)	15 (29%)	21 (41%)	0.1
**Survey completion, n (%)**				
Alone	7 (14%)	7 (14%)	0 (0%)	
With family member	10 (20%)	10 (20%)	0 (0%)	
With healthcare	8 (16%)	8 (16%)	0 (0%)	
No response	26 (51%)	0 (0%)	26 (51%)	

^1^ Pearson’s Chi-squared test; Fisher’s exact test; Wilcoxon rank sum test.

## Data Availability

The data related to this study can be made available upon reasonable request to the authors.

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
