# Peer review of "French Survey on Pain Perception and Management in Patients with Locked-In Syndrome"

_diagnostics, 2022, doi:10.3390/diagnostics12030769_

Round 1
Reviewer 1 Report
Thank you for the opportunity to review this interesting manuscript. The first goal was to confirm or weaken the results obtained in previous studies on the presence of pain and the impact on quality of life. Indeed, since LIS patients often suffer from spasticity and prolonged immobility, hypothesis the chronic pain in these patients is very common, and also that pain will have a negative impact on their quality of life, affecting their mental abilities, sleep and emotions. The second purpose of this survey was to provide more detailed information on the characteristics and treatment of pain in this particular population (type and frequency of pain, ways of communicating pain, type of treatment used, degree of satisfaction with these treatments, etc.). Autors expect most patients to resort to drug treatments, and only a few have tested non-pharmacological treatments such as hypnosis and meditation.
Systematic implementation by clinical teams of a specific pain communication code can help detect these symptoms more quickly.
Conclusion - The possibility of proposing a global non-pharmacological approach such as hypnosis and / or meditation may allow patients to reduce pain while maintaining the level of arousal in order to avoid misdiagnosis due to fatigue - in my opinion this has not been confirmed in the conducted studies, on what basis in of LIS patients were non-pharmacological pain management methods tested? What data presented in the study confirms the effectiveness of the therapy used in patients with LIS?
Author Response
Reviewer 1
Thank you for the opportunity to review this interesting manuscript. The first goal was to confirm or weaken the results obtained in previous studies on the presence of pain and the impact on quality of life. Indeed, since LIS patients often suffer from spasticity and prolonged immobility, hypothesis the chronic pain in these patients is very common, and also that pain will have a negative impact on their quality of life, affecting their mental abilities, sleep and emotions. The second purpose of this survey was to provide more detailed information on the characteristics and treatment of pain in this particular population (type and frequency of pain, ways of communicating pain, type of treatment used, degree of satisfaction with these treatments, etc.). Autors expect most patients to resort to drug treatments, and only a few have tested non-pharmacological treatments such as hypnosis and meditation.
Systematic implementation by clinical teams of a specific pain communication code can help detect these symptoms more quickly. Conclusion - The possibility of proposing a global non-pharmacological approach such as hypnosis and / or meditation may allow patients to reduce pain while maintaining the level of arousal in order to avoid misdiagnosis due to fatigue - in my opinion this has not been confirmed in the conducted studies, on what basis in of LIS patients were non-pharmacological pain management methods tested? What data presented in the study confirms the effectiveness of the therapy used in patients with LIS?
Response: We thank the reviewer for the careful reading of our manuscript and for the relevance of the feedback provided. To clarify some aspects, we would like to point out that in our study only a few participants have tried non-pharmacological treatments such as osteopathy, acupuncture and electromagnetism but none have tested hypnosis or meditation. We expect patients with LIS to benefit from hypnosis and meditation as this has been the case in other clinical populations (Bicego et al., 2021; Deng, 2019; Ngamkham et al., 2019; Vanhaudenhuyse et al., 2018). Previous neuroimaging studies also support this view in which meditation experts have shown decreased pain sensitivity linked to decreased brain activity in the executive and emotion regions. This decrease in the cognitive and emotional control of pain could allow participants to experience pain as a neutral instead of unpleasant stimulation. Regarding the use of hypnosis to modulate pain, studies have also shown a decrease of brain activation in areas involved in sensory and affective processing of pain during hypnosis. This technique has also been shown to be effective in reducing pain in patients with chronic pain. Based on these data, we find it relevant to suggest integrating hypnosis and meditation as non-pharmacological treatments for LIS patients in a larger scale study to eventually confirm our hypotheses.
However, we agree with the reviewer that this discussion belongs to the discussion section and not the conclusion of the study. We have now modified the information in the discussion and conclusion sections.
You can now read in the discussion (lines 374-377): “Therefore, it would be relevant to further investigate the potential benefit of hypnosis and meditation in the management of LIS patients. Proposing a global non-pharmacological approach could help reducing pain while preserving patient’s level of arousal.”
You can now read in the conclusion (lines 419-421) “An exciting other option to reduce pain and avert these side effects would be to use non-pharmacological treatments, but this requires further investigation in future studies.”
Reviewer 2 Report
This is an interesting study of a small sample with the locked-in syndrome. I have a few comments:
- As the authors state, this study is based on a very small study population. I would call it a pilot study. Such a study has the purpose of reporting new findings to be confirmed or not confirmed in a up-coming more large-scale study.
- I lack a description of how the study population was recruited. Okay, 51 subjects responded to the questionnaire. How did you find them? And how many did you find that not responded to the questionnaire? There is a deficiency of study population definition.
- No informed consent was obtained from the participants. That might be OK in a pilot study, but not in a more large-scale study. No matter what Belgian or French law says, scientific international agreements override Belgian and French law. In a coming large-scale study informed consent just is a must.
- Since this might be regarded as a pilot study the text is far too long. Publishers usually claim that text and illustrations, such as text, tables and figures, must not overlap. In this manuscript the content of all tables and all figures are minutely described in the text. My suggestion is that the authors reduce the text describing illustrations and let these stand for them selves.
Author Response
Reviewer 2
This is an interesting study of a small sample with the locked-in syndrome. I have a few comments:
- As the authors state, this study is based on a very small study population. I would call it a pilot study. Such a study has the purpose of reporting new findings to be confirmed or not confirmed in a up-coming more large-scale study.
Response: We thank the reviewer for their interest in our work. The size of our sample can be explained by the fact that LIS is a rare pathology and there are only a few LIS patients. Since its creation 25 years ago, the Association for the LIS has registered in total 916 patients in France and Belgium, of whom approximatively 300 are currently living (as of November 2021). In addition, the vast majority of previous studies in the literature on LIS patients are case studies and the few group studies are about the same size as this study (e.g., Branco et al., 2021: 28 participants; Rousseau et al., 2013: 19 participants; Bruno et al., 2011: 65 participants; Khalili-Ardali et al., 2021: 4 paticipants; Svernling et al., 2019: 14 participants; Corallo et al., 2017: 7 participants; Leonard et al., 2019: 15 participants; Lugo et al., 2015: 88 participants).
We have clarified this point and we have now justified our sample size. However, we definitely agree that it would be interesting to conduct a larger scale study to confirm these results. We had indeed mentioned it in our limitations (lines 383-387): “Firstly, the sample size is small (n=51 in total but only 25 who completed the entire questionnaire). It would be interesting to follow-up with a larger representative sample. Unfortunately, as LIS is a rare pathology gathering a large sample remains a challenging consideration for future studies.”
You can now read in the discussion (lines 387-389): “Nevertheless, the vast majority of studies in the literature on LIS patients are case studies and published group studies are approximately the same size as our sample size [20,43–49].”
- I lack a description of how the study population was recruited. Okay, 51 subjects responded to the questionnaire. How did you find them? And how many did you find that not responded to the questionnaire? There is a deficiency of study population definition.
Response: Thank you for this relevant comment. We emailed participants who were part of the ALIS database (approximatively 300 in the current database). Out of these, 59 completed the questionnaire and 8 were excluded from the analysis (incomplete data). A total of 51 participants were included in the analyses. Thus, the response rate was around 17%. This rate is rather low but it should be put into perspective with the context in which the study was carried out and the type of population targeted. First, we do not know how many participants did actually read the email. Secondly, some LIS patients may not have the tools and help to fill the questionnaire. Third, ALIS sends several questionnaires per year, and this can become discouraging for the LIS patients because it requires time and special effort.
We added this information in the methods (lines 92-93) and results (lines 162-165) as follows: “This survey was sent by email to LIS members of the Association du Locked-In Syndrome (ALIS, Paris, France)”; “From the 300 patients contacted by email, 59 surveys were collected, and 8 were excluded from the analysis as there were incomplete data on pain. A total of 51 participants (41 from the online version and 10 from the paper version) were included in the analysis, representing a 17% acquisition rate.”
And we also added a comment in the limitation section (lines 389-394) as follows: “The response rate however was low (17%) but this should be put into perspective with the context in which the study was carried out and the type of population targeted. Indeed, we do not know how many participants did actually read the email; some LIS patients may not have had the tools and help to fill the questionnaire; and ALIS sends several questionnaires per year, which can be discouraging for the LIS patients because it requires time and effort.”
- No informed consent was obtained from the participants. That might be OK in a pilot study, but not in a more large-scale study. No matter what Belgian or French law says, scientific international agreements override Belgian and French law. In a coming large-scale study informed consent just is a must.
Response: In the event that we conduct a large-scale study, informed consent will be asked specifically. Therefore, we are committed to respect international scientific agreements and obtain informed consent from each participant. Note that there is the assumption that participants who fill the questionnaire give their consent by filling the questionnaire, and this procedure has been accepted by our Ethical Committee Advisory Board. As mentioned in the method section (lines 98-99): “Completion of the questionnaire was non-remunerated, voluntary, and anonymous. Completion was considered as consent for participation in the survey”.
- Since this might be regarded as a pilot study the text is far too long. Publishers usually claim that text and illustrations, such as text, tables and figures, must not overlap. In this manuscript the content of all tables and all figures are minutely described in the text. My suggestion is that the authors reduce the text describing illustrations and let these stand for themselves.
Response: We agree with the reviewer that the manuscript was quite long and we now have shortened it. We mostly reduced the length of the results section by deleting parts of the results that were shown in the figures and we also replaced the text on demographical texts and main characteristics of pain in two summary tables (lines 179 and 194), as requested by another reviewer.
Deleted texts related to figures: Figure 2: “Among the 25 participants who experienced pain, 21 (84%) located their pain in their lower limbs, seven (28%) had headaches, six (24%) had pain in their upper limbs, five (20%) in their back, four (16%) in the abdominals and two (8%) answered that the pain was diffuse (Figure 2 and Supplementary Table S2).”; “Finally, regarding how patients communicate pain, more than half of the participants did not express their pain (13/25; 52%), the others expressed it through cries (11/25; 44%), vocalization (7/25; 28%), by using a communication code (7/25; 28%), winces (6/25; 24%), look/ blinking (4/25; 16%), and two participants used other ways such as verbalization via a speech valve and the presence of acute spasticity (2/25; 8%) (Figure 2C) (Table 2).” Figure 3: “The elements that most increased pain were the supine (11/25; 44%) and sitting positions (11/25; 44%), followed by care (nursing or physiotherapy; 9/25; 36%), tiredness (9/25; 36%), touching (7/25; 28%), temperature (4/25; 16%), mood/emotion (4/25; 16%), equipment (3/25; 12%), physical exercises (2/25; 8%), and other causes, such as feeding (1/25; 4%) and daily handling (1/25; 4%). One participant did not report any factor (1/25; 4%). On the other hand, when patients were asked which elements help to reduce their pain levels, supine position (9/25; 36%) and other causes such as being busy, botox injection, use of medication (i.e., paracetamol), vocalizing pain, mobilization, standing/tilting the back, and massage (9/25; 36%) were the most common, followed by care (5/25; 20%), touching (5/25; 20%), sitting position (4/25; 16%), temperature (3/25; 12%), physical exercises (2/25; 8%), equipment (2/25; 8%), and tiredness (1/25; 4%)” Figure 4: “cognition, only nine patients answered that pain had no effect on their cognitive abilities while for the others, pain seemed to increase mood swings (13/25; 52%), tiredness (14/25; 56%) and depression (6/25, 24%), and decrease concentration/attention (4/25; 16%) and memory capacities (4/25; 16%). Ten out of 25 patients reported that pain disrupted their sleep (46%, mean influence = 6.42 ± 3.23), and the majority of the patients reported that pain sometimes (14/25; 56%) or always (4/25; 16%) have an influence on their emotions (mean influence = 4.68 ± 2.92)” Figure 5: “The majority of the participants responded that they were taking pharmacological treatments (15/25; 60%) and 10 answered that they did not (40%). Among them, participants reported taking pharmacological treatment once a day (8/25; 53%), several times a day (5/25; 33%) or occasionally (2/25; 13%). Participants were also asked about the type of pharmacological treatments they used. Painkillers level 1 were most often used (12/15; 73%), followed by anti-inflammatory (3/15; 20%), painkillers level 2 (2/15; 13%), painkiller level 3 (1/15; 7%), myorelaxant (1/15; 7%), anti-epileptic (1/15; 7%), anti-depressant (1/15; 7%), local anesthetic (1/15; 7%), and benzodiazepine (1/15; 7%). Two participants did not know which type of medication they were taking (13%) (Figure 5A). It should be noted that more than half of these participants used at least two different types of medications (8/15; 53%). According to the patients interviewed, the efficiency of these pharmacological treatments was judged to be moderate (mean efficiency = 6±2.93) (Figure 5B).”; “Only three out 25 (12%) participants responded that they were taking non pharmacological treatments, 21 (84%) said that they did not, and one (4%) did not know. Non-pharmacological treatments included osteopathy, acupuncture and electromagnetism. The efficiency of these non-pharmacological treatments was judged to be low (mean efficiency = 4±1.73). Note that among these three participants, two were also taking pharmacological treatments to manage pain. Finally, 12 (48%) patients said they would be willing to try a new pharmacological treatment compared to only six (24%) who would be willing to try a non-pharmacological treatment such as hypnosis or meditation (Figure 5B, C and D).”
Reviewer 3 Report
The authors proposed a French survey on pain perception and management in patients with locked-in syndrome. The idea is of interest and the authors could receive a promising finding, however, some major points should be addressed:
1. Sample size is too small for a survey study (only 51 participants), thus the results contained some biases.
2. Patient characteristics must contain statistical analyses and p-values.
3. The authors should follow some standard guidelines when reporting the results.
4. English language should be minor checked and fixed.
Author Response
Reviewer 3
The authors proposed a French survey on pain perception and management in patients with locked-in syndrome. The idea is of interest and the authors could receive a promising finding, however, some major points should be addressed:
We thank the reviewer for his/her interest in our study and for the constructive remarks.
- Sample size is too small for a survey study (only 51 participants), thus the results contained some biases.
Response: Indeed, one limitation that we already acknowledged in the manuscript is the small size of our sample which makes our results difficult to generalize.
As mentioned to reviewer 2, the size of our sample can be explained by the fact that LIS is a rare pathology and there are only a few LIS patients. Since its creation 25 years ago, the Association for the LIS has registered in total 916 patients in France and Belgium, of whom approximatively 300 are currently living (as of November 2021). In addition, the vast majority of previous studies in the literature on LIS patients are case studies and the few group studies are about the same size as this study (e.g., Branco et al., 2021: 28 participants; Rousseau et al., 2013: 19 participants; Bruno et al., 2011: 65 participants; Khalili-Ardali et al., 2021: 4 paticipants; Svernling et al., 2019: 14 participants; Corallo et al., 2017 : 7 participants; Leonard et al., 2019: 15 participants ; Lugo et al., 2015: 88 participants).
We have clarified this point and we have now justified our sample size. However, we definitely agree that it would be interesting to conduct a larger scale study to confirm these results. We had indeed mentioned it in our limitations (lines 383-387): “Firstly, the sample size is small (n=51 in total but only 25 who completed the entire questionnaire). It would be interesting to follow-up with a larger representative sample. Unfortunately, as LIS is a rare pathology gathering a large sample remains a challenging consideration for future studies.”
You can now read in the discussion (lines 387-389): “Nevertheless, the vast majority of studies in the literature on LIS patients are case studies and published group studies are approximately the same size as our sample size [20,43–49].”
- Patient characteristics must contain statistical analyses and p-values.
Response: This information is in Tables S3 and S4, and we now refer to these supplementary materials in the main text. We leave it to the editor to move it in the main document or to leave it as supplementary materials.
- The authors should follow some standard guidelines when reporting the results.
Response: Indeed, we have added the CHERRIES checklist as a Supplementary Material and added more details in the method section accordingly.
Lines 92-93: “This survey was sent by email to LIS members of the Association du Locked-In Syndrome (ALIS, Paris, France).”
Lines 97-98: “Completion of the questionnaire was non-remunerated, voluntary, anonymous and was considered as consent for participation in the survey.”
Lines 100-101: “The questionnaire follows the CHERRIES checklist guidelines (see Supplementary Material 1). The survey was developed on the basis of previous surveys carried out among LIS patients studying their well-being and quality of life [20,26] and tested internally by the investigators before sending it to the participants. The online survey was done using a Google form and a paper version was also made available (PDF).”
Lines 151-152: “Data were exported and analyzed from the Google form in .csv format. Data collected from the paper version were added to these files”
- English language should be minor checked and fixed.
We thank the reviewer for this comment. We now asked a native UK speaker to check the manuscript (see changes highlighted in yellow in the manuscript) and we added him in the author list as he also provided useful feedback to improve the manuscript.
Reviewer 4 Report
I liked reading this paper, which presents the results of a survey about pain in LIS patients. It maps, e.g., the types of pain in these patients, which types of treatment they receive, and how they communicate the pain. I found the last point (communication) most important, and the authors also emphasize this. Given the lack of research in this area, the research is valuable. The study does not test any specific hypotheses but is mainly explorative.
I feel that the paper could be shortened, especially in the presentation of the results. I would summarize the tables instead of presenting one row per participant; one or two short tables could summarize all relevant information concisely. E.g., you could report *frequencies* for type of communication/wheelchair/survey completion etc., *average* time since injury + other continuous variables, i.e., data over all participants rather than per individual. I also feel that the figures, although they look nice, are somewhat redundant because they do not convey more information than the text. However, I understand that they can be used to illustrate the key factors that the authors consider as most relevant/important.
I suggest that in future studies the authors would use questions with answers on scales which would enable e.g. correlation or regression analyses to gain better insight into the experience of pain in these patients. E.g., are there some special aspects about LIS patients that could affects their pain? E.g., how they are moved by helpers, or is the communication of pain associated with their pain (maybe this could be tested even with the present data), or other social/functional aspects related to pain experience which differ in LIS compared to others.
Author Response
I liked reading this paper, which presents the results of a survey about pain in LIS patients. It maps, e.g., the types of pain in these patients, which types of treatment they receive, and how they communicate the pain. I found the last point (communication) most important, and the authors also emphasize this. Given the lack of research in this area, the research is valuable. The study does not test any specific hypotheses but is mainly explorative.
Response: We appreciate the words of the reviewer and the importance that he/she gives to our study. We are pleased with the enthusiasm shown when reading our article. We are also grateful for the relevant comments that we took into account.
I feel that the paper could be shortened, especially in the presentation of the results. I would summarize the tables instead of presenting one row per participant; one or two short tables could summarize all relevant information concisely. E.g., you could report *frequencies* for type of communication/wheelchair/survey completion etc., *average* time since injury + other continuous variables, i.e., data over all participants rather than per individual. I also feel that the figures, although they look nice, are somewhat redundant because they do not convey more information than the text. However, I understand that they can be used to illustrate the key factors that the authors consider as most relevant/important.
Response: The paper has now been shortened, as also suggested by another reviewer and we have added summarized tables. However, we would like to keep the individual data in the supplementary materials in case someone would like to do a meta-analysis, so that they have the needed information.
We have deleted the parts of the results section related to the figures: Figure 2: “Among the 25 participants who experienced pain, 21 (84%) located their pain in their lower limbs, seven (28%) had headaches, six (24%) had pain in their upper limbs, five (20%) in their back, four (16%) in the abdominals and two (8%) answered that the pain was diffuse (Figure 2 and Supplementary Table S2).”; “Finally, regarding how patients communicate pain, more than half of the participants did not express their pain (13/25; 52%), the others expressed it through cries (11/25; 44%), vocalization (7/25; 28%), by using a communication code (7/25; 28%), winces (6/25; 24%), look/ blinking (4/25; 16%), and two participants used other ways such as verbalization via a speech valve and the presence of acute spasticity (2/25; 8%) (Figure 2C) (Table 2).”Figure 3: “The elements that most increased pain were the supine (11/25; 44%) and sitting positions (11/25; 44%), followed by care (nursing or physiotherapy; 9/25; 36%), tiredness (9/25; 36%), touching (7/25; 28%), temperature (4/25; 16%), mood/emotion (4/25; 16%), equipment (3/25; 12%), physical exercises (2/25; 8%), and other causes, such as feeding (1/25; 4%) and daily handling (1/25; 4%). One participant did not report any factor (1/25; 4%). On the other hand, when patients were asked which elements help to reduce their pain levels, supine position (9/25; 36%) and other causes such as being busy, botox injection, use of medication (i.e., paracetamol), vocalizing pain, mobilization, standing/tilting the back, and massage (9/25; 36%) were the most common, followed by care (5/25; 20%), touching (5/25; 20%), sitting position (4/25; 16%), temperature (3/25; 12%), physical exercises (2/25; 8%), equipment (2/25; 8%), and tiredness (1/25; 4%)” Figure 4: “cognition, only nine patients answered that pain had no effect on their cognitive abilities while for the others, pain seemed to increase mood swings (13/25; 52%), tiredness (14/25; 56%) and depression (6/25, 24%), and decrease concentration/attention (4/25; 16%) and memory capacities (4/25; 16%). Ten out of 25 patients reported that pain disrupted their sleep (46%, mean influence = 6.42 ± 3.23), and the majority of the patients reported that pain sometimes (14/25; 56%) or always (4/25; 16%) have an influence on their emotions (mean influence = 4.68 ± 2.92)” Figure 5: “The majority of the participants responded that they were taking pharmacological treatments (15/25; 60%) and 10 answered that they did not (40%). Among them, participants reported taking pharmacological treatment once a day (8/25; 53%), several times a day (5/25; 33%) or occasionally (2/25; 13%). Participants were also asked about the type of pharmacological treatments they used. Painkillers level 1 were most often used (12/15; 73%), followed by anti-inflammatory (3/15; 20%), painkillers level 2 (2/15; 13%), painkiller level 3 (1/15; 7%), myorelaxant (1/15; 7%), anti-epileptic (1/15; 7%), anti-depressant (1/15; 7%), local anesthetic (1/15; 7%), and benzodiazepine (1/15; 7%). Two participants did not know which type of medication they were taking (13%) (Figure 5A). It should be noted that more than half of these participants used at least two different types of medications (8/15; 53%). According to the patients interviewed, the efficiency of these pharmacological treatments was judged to be moderate (mean efficiency = 6±2.93) (Figure 5B).”; “Only three out 25 (12%) participants responded that they were taking non pharmacological treatments, 21 (84%) said that they did not, and one (4%) did not know. Non-pharmacological treatments included osteopathy, acupuncture and electromagnetism. The efficiency of these non-pharmacological treatments was judged to be low (mean efficiency = 4±1.73). Note that among these three participants, two were also taking pharmacological treatments to manage pain. Finally, 12 (48%) patients said they would be willing to try a new pharmacological treatment compared to only six (24%) who would be willing to try a non-pharmacological treatment such as hypnosis or meditation (Figure 5B, C and D).”
New tables:
Table 1: Summary of the socio-demographic information and clinical status of patients included in the study.
|
Variables |
Effective (%) |
Variables |
Effective (%) |
|
|
Sex |
51 (100) |
Gastrostomy |
51 (100) |
|
|
Female |
18 (35) |
No |
17 (34) |
|
|
Male |
18 (35) |
Yes |
34 (66) |
|
|
Unknown |
15 (30) |
Verbal communication |
51 (100) |
|
|
Time since injury |
51 (100) |
No |
38 (74) |
|
|
Within the last 10 years |
26 (51) |
Yes |
13 (26) |
|
|
For 10 years or more |
14 (27) |
Use of an alphabetic code |
51 (100) |
|
|
For 20 years or more |
11 (22) |
No |
16 (31) |
|
|
Etiology |
51 (100) |
Yes |
35 (69) |
|
|
Infection |
2 (4) |
Own a wheelchair |
51 (100) |
|
|
Stroke |
40 (78) |
No |
15 (29) |
|
|
Traumatic brain injury |
4 (8) |
Yes |
36 (71) |
|
|
Aneurysm |
1 (2) |
Survey completion |
51 (100) |
|
|
Other (e.g., meningioma) |
4 (8) |
Alone |
7 (14) |
|
|
Tracheotomy |
51 (100) |
With family member |
10 (19) |
|
|
No |
25 (49) |
With healthcare professional |
8 (16) |
|
|
Yes |
26 (51) |
Not specified |
26 (51) |
Table 2: Summary of the main characteristics of pain in LIS patients included in the survey (WHO = World Health Organisation).
|
Variables |
Effective (%) |
Variables |
Effective (%) |
|
|
Pain localization |
25 |
Pain communication |
25 |
|
|
Abdominal |
4 (16) |
Communication code |
7 (28) |
|
|
Lower limb |
21 (84) |
Winces |
6 (24) |
|
|
Upper limb |
6 (24) |
Cries |
11 (44) |
|
|
Head |
7 (28) |
Look/blinking |
4 (16) |
|
|
Back |
5 (20) |
Vocalizations |
7 (28) |
|
|
Diffuse |
2 (8) |
Spasticity |
1 (4) |
|
|
Pain features |
25 |
Verbalization |
1 (4) |
|
|
Electrical shocks |
10 (40) |
No expression of pain |
13 (52) |
|
|
Burn |
6 (24) |
Pharmacological treatments |
25 |
|
|
Painful cold sensation |
4 (16) |
None |
10 (40) |
|
|
No features |
10 (40) |
Yes |
15 (60) |
|
|
Time since pain |
25 |
Type of pharmacological treatments |
15 |
|
|
1-3 months |
2 (8) |
Painkiller level 1 |
12 (73) |
|
|
3-6 months |
6 (24) |
Painkiller level 2 |
2 (13) |
|
|
6 months - one year |
5 (20) |
Painkillers level 3 |
1 (7) |
|
|
> one year |
12 (48) |
Anti-epileptic |
1 (7) |
|
|
Pain frequency |
25 |
Anti-depressant |
1 (7) |
|
|
Discrete pain |
21 (84) |
Anti-inflammatory |
3 (20) |
|
|
Continuous pain |
4 (16) |
Myorelaxant |
1 (7) |
|
|
Discrete pain |
21 |
Local anesthetic |
1 (7) |
|
|
> once a day |
4 (19) |
Benzodiazepine |
1 (7) |
|
|
Once a day |
2 (10) |
|
|
|
|
< once a day |
3 (14) |
|
|
|
|
Unknow |
12 (57) |
|
|
I suggest that in future studies the authors would use questions with answers on scales which would enable e.g. correlation or regression analyses to gain better insight into the experience of pain in these patients. E.g., are there some special aspects about LIS patients that could affects their pain? E.g., how they are moved by helpers, or is the communication of pain associated with their pain (maybe this could be tested even with the present data), or other social/functional aspects related to pain experience which differ in LIS compared to others.
Response: We thank the reviewer for this valuable comment. LIS patients represent a particular population with reduced means of communication. We therefore wanted to simplify their task by proposing binary answers (e.g., yes/no), which constitutes a basic communication code. Filling out a questionnaire already requires considerable effort for them, so we wanted to make the task as easy as possible. In addition, the questionnaire was already long so we did not want to overload the participants. However, we agree that in future studies the use of questions with multiple answers on Likert scales should be considered to collect more detailed data.
You can now read in the discussion (lines 405-407): “Additionally, future studies should use alternative response format (e.g., Likert scales) to collect more detailed data than binary format (e.g., yes/no).”
References
Bicego, A., Monseur, J., Collinet, A., Donneau, A.-F., Fontaine, R., Libbrecht, D., Malaise, N., Nyssen, A.-S., Raaf, M., Rousseaux, F., Salamun, I., Staquet, C., Teuwis, S., Tomasella, M., Faymonville, M.-E., & Vanhaudenhuyse, A. (2021). Complementary treatment comparison for chronic pain management: A randomized longitudinal study. PLOS ONE, 16(8), 1–20. https://doi.org/10.1371/journal.pone.0256001
Branco, M. P., Pels, E. G. M., Sars, R. H., Aarnoutse, E. J., Ramsey, N. F., Vansteensel, M. J., & Nijboer, F. (2021). Brain-Computer Interfaces for Communication: Preferences of Individuals With Locked-in Syndrome. Neurorehabilitation and Neural Repair, 35(3), 267–279. https://doi.org/10.1177/1545968321989331
Bruno, M.-A., Bernheim, J. L., Ledoux, D., Pellas, F., Demertzi, A., & Laureys, S. (2011). A survey on self-assessed well-being in a cohort of chronic locked-in syndrome patients: Happy majority, miserable minority. BMJ Open, 1(1), e000039–e000039. https://doi.org/10.1136/bmjopen-2010-000039
Corallo, F., Bonanno, L., Lo Buono, V., De Salvo, S., Rifici, C., Pollicino, P., Allone, C., Palmeri, R., Todaro, A., Alagna, A., Bramanti, A., Bramanti, P., & Marino, S. (2017). Augmentative and Alternative Communication Effects on Quality of Life in Patients with Locked-in Syndrome and Their Caregivers. Journal of Stroke and Cerebrovascular Diseases, 26(9), 1929–1933. https://doi.org/10.1016/j.jstrokecerebrovasdis.2017.06.026
Deng, G. (2019). Integrative Medicine Therapies for Pain Management in Cancer Patients. The Cancer Journal, 25(5), 343–348. https://doi.org/10.1097/PPO.0000000000000399
Khalili-Ardali, M., Wu, S., Tonin, A., Birbaumer, N., & Chaudhary, U. (2021). Neurophysiological aspects of the completely locked-in syndrome in patients with advanced amyotrophic lateral sclerosis. Clinical Neurophysiology, 132(5), 1064–1076. https://doi.org/10.1016/j.clinph.2021.01.013
Leonard, M., Renard, F., Harsan, L., Pottecher, J., Braun, M., Schneider, F., Froehlig, P., Blanc, F., Roquet, D., Achard, S., Meyer, N., & Kremer, S. (2019). Diffusion tensor imaging reveals diffuse white matter injuries in locked-in syndrome patients. PLOS ONE, 14(4), e0213528. https://doi.org/10.1371/journal.pone.0213528
Lugo, Z. R., Bruno, M.-A., Gosseries, O., Demertzi, A., Heine, L., Thonnard, M., Blandin, V., Pellas, F., & Laureys, S. (2015). Beyond the gaze: Communicating in chronic locked-in syndrome. Brain Injury, 29(9), 1056–1061. https://doi.org/10.3109/02699052.2015.1004750
Ngamkham, S., Holden, J. E., & Smith, E. L. (2019). A Systematic Review: Mindfulness Intervention for Cancer-Related Pain. Asia-Pacific Journal of Oncology Nursing, 6(2), 161–169. https://doi.org/10.4103/apjon.apjon_67_18
Rousseau, M.-C., Pietra, S., Nadji, M., & Billette de Villemeur, T. (2013). Evaluation of Quality of Life in Complete Locked-In Syndrome Patients. Journal of Palliative Medicine, 16(11), 1455–1458. https://doi.org/10.1089/jpm.2013.0120
Svernling, K., Törnbom, M., Nordin, Å., & Sunnerhagen, K. S. (2019). Locked-in syndrome in Sweden, an explorative study of persons who underwent rehabilitation: A cohort study. BMJ Open, 9(4), e023185. https://doi.org/10.1136/bmjopen-2018-023185
Vanhaudenhuyse, A., Gillet, A., Malaise, N., Salamun, I., Grosdent, S., Maquet, D., Nyssen, A.-S., & Faymonville, M.-E. (2018). Psychological interventions influence patients’ attitudes and beliefs about their chronic pain. Journal of Traditional and Complementary Medicine, 8(2), 296–302. https://doi.org/10.1016/j.jtcme.2016.09.001
Round 2
Reviewer 3 Report
My previous comments have been addressed.
Author Response
Thank you very much for your comments.